# eXIAA: eXplainable Injections for Adversarial Attack

## Abstract

Post-hoc explainability methods are a subset of Machine Learning (ML) that aim to provide a reason for why a model behaves in a certain way. In this paper, we show a new black-box model-agnostic adversarial attack for post-hoc explainable Artificial Intelligence (XAI), particularly in the image domain. The goal of the attack is to modify the original explanations while being undetected by the human eye and maintain the same predicted class. In contrast to previous methods, we do not require any access to the model or its weights, but only to the model's computed predictions and explanations. Additionally, the attack is accomplished in a single step while significantly changing the provided explanations, as demonstrated by empirical evaluation. The low requirements of our method expose a critical vulnerability in current explainability methods, raising concerns about their reliability in safety-critical applications. We systematically generate attacks based on the explanations generated by post-hoc explainability methods (saliency maps, integrated gradients, and DeepLIFT SHAP) for pretrained ResNet-18 and ViT-B16 on ImageNet. The results show that our attacks could lead to dramatically different explanations without changing the predictive probabilities. We validate the effectiveness of our attack, compute the induced change based on the explanation with mean absolute difference, and verify the closeness of the original image and the corrupted one with the Structural Similarity Index Measure (SSIM).

## 1 Introduction

As deep learning models have become more complex and applied across various fields, the demand for transparency in artificial intelligence (AI) outcome has also accordingly increased, particularly in some scientific domains (Lipton, 2018; Mengaldo, 2024; Imrie et al., 2023). Nowadays, explaining why a model makes certain decisions is as important as prediction itself. Being unable to explain an output is detrimental to the widespread adoption of AI and the user trust (Shin, 2021). Furthermore, in high-risk sectors, like medicine and healthcare (Chaddad et al., 2023), autonomous driving and industry automation (Atakishiyev et al., 2024), and finance (Weber et al., 2024), providing explanations is required for safety concerns, beyond just making accurate predictions. Many explainable artificial intelligence (XAI) methods, especially feature attribution approaches, have been developed to pinpoint input features that significantly influence the model outcome. They are often categorized into ante-hoc methods (interpretable models) (Turbé et al., 2024; Li et al., 2018), which are model-specific and produce an explanation along with the prediction, and post-hoc methods (Turbé et al., 2023; Samek et al., 2017), which are model-agnostic and can be applied to a wide variety of models. For an in-depth overview of explainability, we refer to the following surveys (Zhang et al., 2021; Samek et al., 2021).

However, having an explanation is not enough. The explanation itself must be reliable and robust to establish trust and transparency between users and the model. For example, clinicians may require the feature attribution details when an AI system assists in diagnosing a disease. Yet if the patient data is maliciously corrupted–nearly indistinguishable from the original and also predicted as a certain disease–but produces a markedly different explanation, it would mislead the clinician towards recognizing a different cause for the disease and, therefore, prescribing the wrong treatment. In such cases, though the model robustly makes correct predictions, fragile explanations pose significant risks for real-world deployment.

Since Ghorbani et al. (2019) first introduced the notion of adversarial perturbations to neural network interpretation, a new category of adversarial attacks has emerged in the domain that target explanations while keeping the model prediction unchanged. These attacks, based on the scenarios, could prove even more dangerous than traditional adversarial attacks. Most of existing adversarial attacks on explanations have highlighted how destructive these attacks can be (Ghorbani et al., 2019; Kindermans et al., 2019; Adebayo et al., 2018; Dombrowski et al., 2019). Despite this, they have not received enough attention due to their impracticality, as they have stringent requirements like access to the model, the ability to modify the model's weights, and the possibility of performing multi-step attacks.

**Our contributions.** In this paper, we propose a novel black-box model-agnostic adversarial attack method that has fewer realistic requirements. Specifically, we generate the attack to the original image by leveraging the explanation of an attack image from a running-up class. We systematically investigate three post-hoc interpretability methods (this includes saliency maps, integrated gradients, and DeepLIFT SHAP) on ImageNet in two different network architectures. We show how to design adversarial perturbation that can disrupt the explanation of the explainability method while being visually undetectable and without significantly altering the classification prediction (Figure 1). Unlike previous work, our approach does not require access to the model architecture, weights, or the ability to modify them, and provides one-step attack by exploiting the explanations of the other classes. These characteristics make the attack far more feasible in real case scenarios, proving how dangerous it could be to rely blindly on AI explanations.

While we focus on the image data in this work because most of explainability methods have been motivated in computer vision domain, the vulnerability of neural network interpretability could be a much broader problem. Our proposed adversarial attack approach could also be applied to other types of data, such as time series, texts, tabular data.

## 2 RELATED WORKS AND PRELIMINARIES

**Related works** Adversarial attacks usually involve a maliciously intention attacker whose goal is to disrupt the performance of deep neural networks. It is a fairly well-studied domain, both from a cybersecurity perspective to safeguard systems and infrastructures, as well as from a AI standpoint, where adversarial attacks can be used to augment datasets and create a more robust and resilient model. Some of the most famous works on adversarial attack include Goodfellow et al. (2014), where they applied a small perturbation in the direction of the gradient of the loss with respect to the input. This provides a one-step attack that proves very effective but requires the ability to perform backpropagation. Madry et al. (2017) improved it by finetuning the attack through iterations, and Carlini & Wagner (2017) formulated the attack as an optimization problem by minimizing the perturbation norm subject to misclassifications. Some work suggests that the model that are robust to adversarial attacks could improve the explainability (Ross & Doshi-Velez, 2018; Dong et al., 2017). The focus of these work is on adversarial attacks on predictions rather than attacks on explanations.

As XAI gained popularity, a natural research direction has been to apply adversarial attacks to explanations. Ghorbani et al. (2019) was one of the first to show the fragility of explanations to small perturbations, Kindermans et al. (2019) added a constant shift to the input image, which, by construction, is compensated by the bias of a neural network. Adebayo et al. (2018) changes the explanations by randomizing the weights of the classification network, and Dombrowski et al. (2019) provided a method to manipulate attack while penalizing the change in model prediction arbitrarily. For more related works, refer to the survey Baniecki & Biecek (2024). All these methods rely on stringent requirements like access to the model weights, the ability to manipulate them, and the possibility of performing an attack in multiple steps.

**Post-hoc feature attribution methods** This class of explainability methods explains model output in terms of the important features of the input. Given the input sample $x_i$ and the model's prediction $f(x_i)$, feature attribution methods seek to pinpoint the portions of the input data that significantly affect the prediction. In doing so, these methods assign feature relevance score to each input feature. These are some of the most popular in the XAI domain, and to our knowledge, no method in this subfield is inherently safe from the attack framework we proposed. We summarize three feature

attribution methods used in this paper below, denoted by $E(\boldsymbol{x}_i; f(\boldsymbol{x}_i))$. All three methods are used from the Captum Python library (Kokhlikyan et al., 2020).

- Saliency maps (Simonyan et al., 2013) is one of the earliest and easiest methods for computing the gradient of the model output with respect to the input. It can be understood as taking a first-order Taylor expansion of the network at the input, and the gradients are coefficients. The absolute value of these coefficients can be taken to represent feature importance, $E(\boldsymbol{x}_i; f(\boldsymbol{x}_i)) = \left| \frac{\partial f(\boldsymbol{x}_i)}{\partial \boldsymbol{x}_i} \right|$.

- Integrated gradients (Sundararajan et al., 2017) improved on some limitations of saliency maps–saturation problem discussed in (Shrikumar et al., 2017; Sundararajan et al., 2017)–by integrating the gradients with the use of a baseline input $\boldsymbol{x}^0$. Letting $\Delta \boldsymbol{x}_i = \boldsymbol{x}_i - \boldsymbol{x}^0$, the feature importance score can be calculated by, $E(\boldsymbol{x}_i; f(\boldsymbol{x}_i)) = \Delta \boldsymbol{x}_i \times \int_{\beta=0}^{1} \frac{\partial f(\boldsymbol{x}^0 + \beta \Delta \boldsymbol{x}_i)}{\partial \boldsymbol{x}_i} d\beta$, where $\beta$ is the scaling coefficient.

- DeepLIFT SHAP (Lundberg & Lee, 2017), by extending DeepLIFT (Shrikumar et al., 2017) to approximate the Shapley value (Shapley et al., 1953). DeepLIFT SHAP takes a distribution of baselines and computes the DeepLIFT attribution for each input-baseline pair and averages the resulting attributions per input, propagating the contribution through the network relative to a reference input.

**Metrics for measuring sample similarity** We use Structural Similarity Index Measure (SSIM) (Wang et al., 2004) to quantify the similarity between two images $x$ and $y$, as follows,

$$\text{SSIM}(x, y) = \frac{(2\mu_x \mu_y + c_1)(2\sigma_{xy} + c_2)}{(\mu_x^2 + \mu_y^2 + c_1)(\sigma_x^2 + \sigma_y^2 + c_2)} \tag{1}$$

where $\mu_x$ and $\mu_y$ are the pixel sample mean, $\sigma_x^2$ and $\sigma_y^2$ are sample variance, $\sigma_{xy}$ is the sample covariance of $x$ and $y$, $c_1 = (k_1 L)^2$ and $c_2 = (k_2 L)^2$ are two variables to stabilize the division with weak denominator, $L$ is the dynamic range of the pixel-values (typically this is $2^{\#\text{bits per pixel}} - 1$), $k_1 = 0.01$ and $k_2 = 0.03$ by default.

## 3 PROPOSED METHODOLOGY

**Problem statement** We consider an image classification problem for a dataset $\mathcal{X}$ where each element of it is a tuple $(\boldsymbol{x}_i, y_i)$, where $\boldsymbol{x}_i \in \mathbb{R}^{W \times H \times C}$ defines the $i$-th image (with width $W \in \mathbb{N}$, height $H \in \mathbb{N}$, and channel $C \in \mathbb{N}$), and $y_i \in \mathbb{R}^J$ defines the $i$-th one-hot encoding of the classification label, where $J$ is the total number of classes. A trained neural network model $f(\cdot) : \mathbb{R}^{W \times H \times C} \to \mathbb{R}^J$ can classify the input image and produce the predicted probability $\hat{y}_i = f(\boldsymbol{x}_i)$. For a given classifier $f(\cdot)$ and an *original image* $\boldsymbol{x}_i$, a post-hoc explainability method $E(\boldsymbol{x}_i; f(\boldsymbol{x}_i) : \mathbb{R}^{W \times H \times C} \to \mathbb{R}^{W \times H \times C}$ produces an explanation $E(\boldsymbol{x}_i; f(\cdot)) = \boldsymbol{z}_i$. Our goal is to devise a visually imperceptible adversarial attack that maximizes the difference in the explanation of the input,

$$\max \sum_{W,H,C} |E(\boldsymbol{x}_i; f(\boldsymbol{x}_i)) - E(\hat{\boldsymbol{x}}_i; f(\hat{\boldsymbol{x}}_i))| = \max \sum_{W,H,C} |\boldsymbol{z}_i - \hat{\boldsymbol{z}}_i| \tag{2}$$

where the sum over $W$, $H$, and $C$ represents the sum over all the pixels of the image, and $\hat{\boldsymbol{z}}_i$ is the explanation of the *corrupted image* $\hat{\boldsymbol{x}}_i$. At the same time, for a successful attack, we aim at minimizing the changes in the probability of the predicted class, such that

$$\min |f(\boldsymbol{x}_i) - f(\hat{\boldsymbol{x}}_i)|, \tag{3}$$

and the visual difference between the original image and the corrupted image as

$$\max \mathcal{D}_{SSIM}(\boldsymbol{x}_i, \hat{\boldsymbol{x}}_i), \tag{4}$$

where $\mathcal{D}_{SSIM}(\cdot, \cdot)$ represents the Structural Similarity Index Measure (SSIM) in Equation 1, a perceptual metric that compares two images on luminance, contrast, and structural similarity in local windows. SSIM ranges between $[-1, 1]$ (most of the time clipped between $[0, 1]$), and is 1 for identical images (hence the maximization), while it is $-1$ (or 0) when there is no structural similarity.

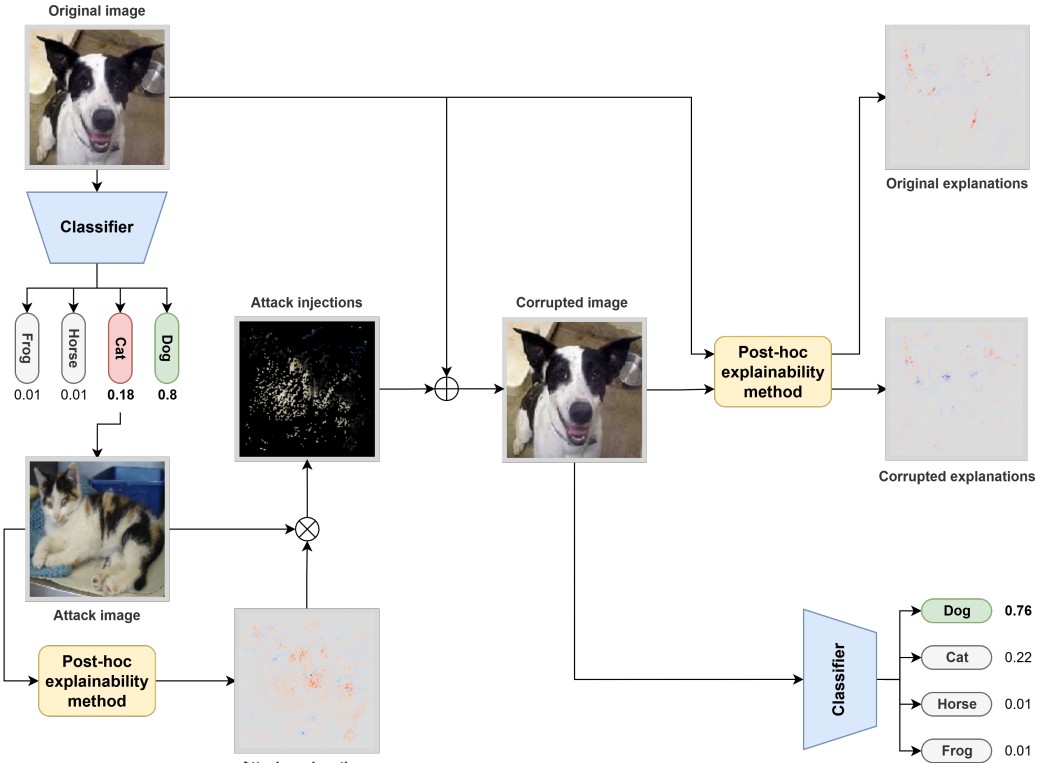

Figure 1: A scheme representing the structure of our adversarial attack on explanations. The classifications' probabilities of the *original image* (dog) are computed through the classifier, from here an *attack image*(cat) from the running-up class is selected. The same post-hoc explainability method of the *original image* is used to extract the positive attribution of the *attack image*. With the top-$k$ positive features selected, we mask the *attack image* ($\otimes$ symbol) and obtain the *attack injections* which are combined with the *original image* with an $\alpha$-weighted sum ($\oplus$ symbol) to create the *corrupted image*. The final corrupted image is not distinguishable from the original one by the human eye, but leads to different explanations.

The maximization in Equation 4 aims to ensure a perturbation that human observers will not easily detect. In Equation 2 and 3, we use absolute difference to measure the changes in the explanations and predictive probabilities. The higher the absolute difference, the greater the difference.

We now discuss how our method generates a corrupted image $\hat{x}$ starting from an original image $x$. For simplicity, we omit the subscript $i$ when referring to the $i$-th image. The main intuition lies in injecting a curated undetectable perturbation that confuses the explainability methods while not altering the classification model prediction. In sum, this paper advocates selecting an image that mostly confuses the classifier to devise an adversarial attack, which is depicted in Figure 1. The proposed method features a three-phase technical pipeline: attack image selection (Sec. 3.1), feature extraction (Sec. 3.2), and feature injection (Sec. 3.3).

## 3.1 PHASE ONE: ATTACK IMAGE SELECTION

In the first phase, we aim to select an *attack image* $\bar{x}$ that will be used to generate an adversarial attack on the original image. As highlighted in several studies on the robustness of the explainability methods (Alvarez-Melis & Jaakkola, 2018; Hooker et al., 2019; Wei et al., 2024), there often is a correlation between the confidence of model predictions and the explanations provided. Therefore, we want to slightly stir the model prediction towards the class with which the model is mostly confused (running-up class). To choose an image, we pass the original image through the classifier and extract the predicted class $y^* = \arg\max_{j \in \{1, \dots, J\}} f_j(x)$, where $f_j(\cdot)$ stands for the $j$-th

classification probability of $y = f(x)$. We extract the running-up class such that

$$y_{\mathrm{r}} = \arg \max_{j \neq y^*} f_j(\boldsymbol{x}).$$

Now we select the image with the highest confidence for the running-up class $y_{\mathrm{r}}$ as follows,

$$\bar{\boldsymbol{x}} = \arg \max_{\boldsymbol{x}' \in \mathcal{X}_{y_{\mathrm{r}}}} f_{y_{\mathrm{r}}}(\boldsymbol{x}').$$

For example, as shown in Figure 1, given the original image (dog), we select the cat as the running-up class, which has the second-highest predictive probability after the dog. The attack image (cat) is the image with the highest confidence in the cat's class. This choice is again motivated by the correlation between high-confidence predictions and strong attribution by the XAI method.

### 3.2 PHASE TWO: FEATURE EXTRACTION

Given the attack image $\bar{\boldsymbol{x}}$, we now want to extract a small but significant set of features in this phase. The selection of a small number of pixels is crucial, as injecting more pixels would make the perturbation more susceptible to being detected. By using the same explainability method $E(\cdot)$ as the original image, we can extract a set of attributions that are relevant to the model and will interfere with the original explanation, once injected into the image. The attributions uncovered not only rely on the color of the pixels' channel but also on the spatial structure between them. We do so by computing the explanation $\bar{\boldsymbol{z}} = E(\bar{\boldsymbol{x}}, f(\cdot))$ and selecting the top-$k$ positive attribution indices as follows,

$$\mathcal{I}_k = \arg \max_{\substack{\mathcal{I} \subseteq [W] \times [H] \times [C] \\ |\mathcal{I}| = k}} \sum_{(w,h,c) \in \mathcal{I}} \bar{\boldsymbol{z}}_{w,h,c},$$

where $[W] = \{1, ..., W\}$, $[C] = \{1, ..., C\}$, $[H] = \{1, ..., H\}$, $w \in [W]$, $h \in [H]$, and $c \in [C]$. The notation $\bar{\boldsymbol{z}}_{w,h,c}$ indicates the value of the attribution at coordinates $w, h, c$. By taking the arguments that maximize the sum over all the attributions and limiting the set $I_k$ to a maximum of $k$ arguments, we extracted the set of $k$ most prominent features according to the post-hoc explainability method.

### 3.3 PHASE THREE: FEATURE INJECTION

We now blend the original image channels with the attack image. Simply substituting the value of the original pixel's channel with the extracted features would drastically change the original image, making the perturbation easily detectable and altering the prediction of the classifier. We use a weighted sum with parameter $\alpha \in (0, 1)$ (to avoid a discoloration, we do so only for the channels extracted), as follows:

$$\hat{\boldsymbol{x}} = \begin{cases} clip((1-\alpha)\boldsymbol{x}_{w,h,c} + \alpha \bar{\boldsymbol{x}}_{w,h,c}) & \text{if } (w, h, c) \in \mathcal{I}_k \\ x_{w,h,c} & \text{otherwise} \end{cases} \tag{5}$$

where $clip$ is a clipping function to the original image domain. The final image $\hat{\boldsymbol{x}}$ is a corrupted version of the original $\boldsymbol{x}$ that is visually similar to it $\mathcal{D}_{SSIM}(\hat{\boldsymbol{x}}, \boldsymbol{x}) \approx 1$, has a close prediction confidence under the classifier $f(\hat{\boldsymbol{x}}) \approx f(\boldsymbol{x})$, but different explanations $E(\hat{\boldsymbol{x}}, f(\hat{\boldsymbol{x}})) \neq E(\boldsymbol{x}, f(\boldsymbol{x}))$.

## 4 EXPERIMENTS

**Datasets and models** For generating attacks against feature attribution explanations, we use ImageNet (Deng et al., 2009). Additionally, we run some secondary results, impractical to compute on ImageNet, on CIFAR-10 (Krizhevsky et al., 2009). For the classification task, we use a pre-trained ResNet-18 model (He et al., 2016) and a ViT-B16 model (Dosovitskiy et al., 2020). All the results are examined on feature attribution explanations obtained by saliency maps, integrated gradients, and DeepLIFT SHAP. We run our attack algorithm for different $\alpha \in [0.03, 0.06, 0.09, 0.12, 0.15]$ and top-$k \in [0.01, 0.05, 0.1, 0.15, 0.2, 0.3, 0.4, 0.5, 0.6, 0.8]$. We choose these values of $\alpha$ to have a broad range of parameters and see how the method performs from being unrecognizable to being quite visible (the notion of visibility differs not only between pairs of original and injection images, but also from human to human, so we opted for a broader range to have a better understanding of the method's performance). Similarly, we choose an irregularly spaced range of top-$k$ that spans from

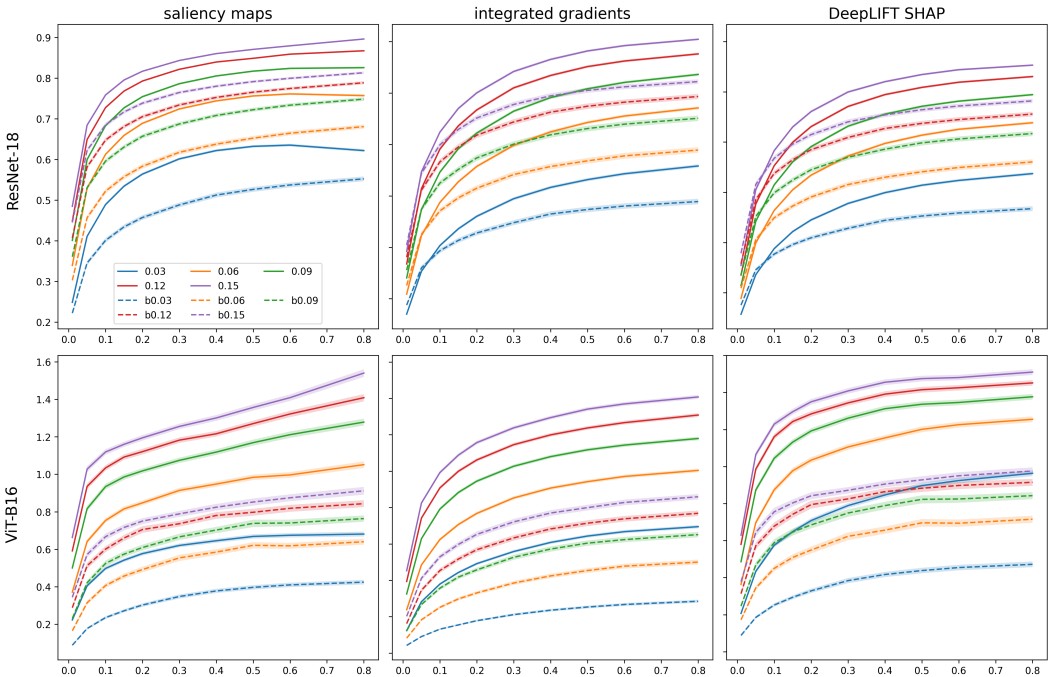

Figure 2: Percentage change of explanations with different $\alpha$. Each graph represents a pair of classifier and explainability method. The x-axis represents the top-$k$ while the y-axis represents the change in explanations. The graphs in the same row share the same y-axis scale. Each line represents the mean and standard deviation of a different value of $\alpha$ as represented in the legend, and corresponding dotted line represents the baseline performance relative to the same $\alpha$.

inserting very few top positive features (hardly detected) to almost all the positive features (more easily detected). The irregularity better showcases the lower part of top-$k$, which is of more interest for our attack and is more dynamic according to our results. We use values of top-$k \in (0, 1)$ to represent the percentage amount of positive features to extract.

We select a random image from a subset of 500 classes. For every image that we attack, we select the 3 images from the running-up class with the highest classification score for that class, and for each of them, we compute the attack with all the possible combinations of the sets of predefined $\alpha$ and top-$k$. Additionally, we compared our method with a baseline, computed by adding Gaussian noise to the original image.

**Explainability performance** Figure 2 represents the percentage change of explanations, that is the difference in explanations between the original and corrupted image (summed across all pixels and channels in absolute values) and divided by the explanations of the original image (this is no different than the sum of absolute difference only reported in a more meaningful scale in reference to the attacked image). The higher the value, the bigger the change in the attacked explanations; a value of $0\%$ means the two explanations are identical. We see that for equal values of $\alpha$ our method significantly outperforms the baseline. Our method proves to be more effective for transformer-based architecture. We see recurring patterns as $\alpha$ and top-$k$ change across different architectures and explainability methods. Consistent with the theory of our approach, as $\alpha$ and top-$k$ increase, so does performance. Nonetheless, $\alpha$ experiences diminishing returns (e.g., the distance between the gap with $\alpha = 0.03$ and $\alpha = 0.06$ is bigger than the one between $\alpha = 0.06$ and $\alpha = 0.09$ and so on). Similarly, for top-$k$, there's a fast improvement at the beginning, which plateaus. This is explained by the meaning of a higher top-$k$, as it approaches 1, the injected features are less and less representative, and small improvements are justified by the additional corrupted pixels. Furthermore, the values $\alpha$ and top-$k$ are dependent on each other. A bigger value of $\alpha$ pushes forward the point from where a higher top-$k$ relates to diminishing improvement.

**Detectability performance** Figure 3 shows for each test how similar the computed corrupted image

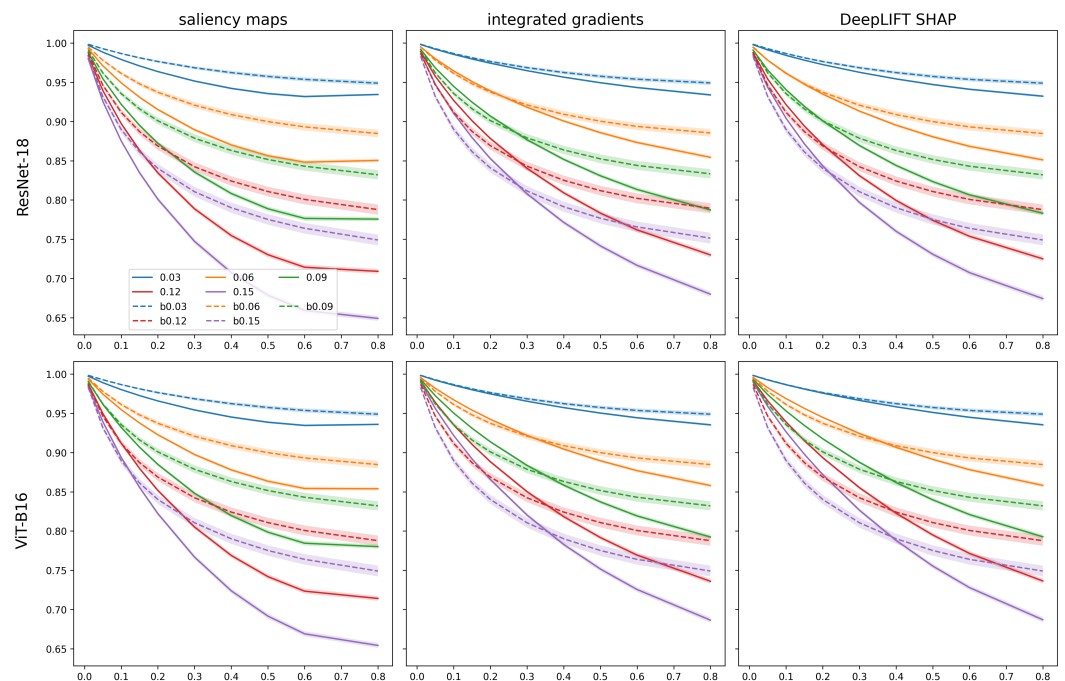

Figure 3: SSIM between the original image and the corrupted one. The structure follows the same one as Figure 2.

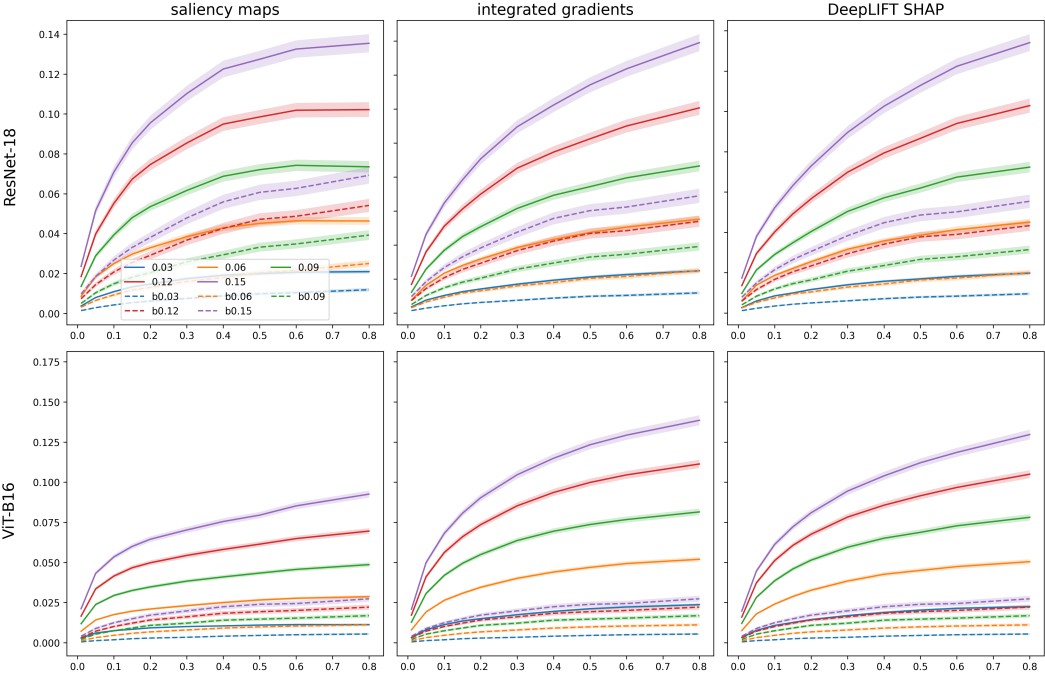

Figure 4: The confidence absolute change for the predicted class of the original image vs the corrupted image. The structure follows the same one as Figure 2.

is to the original image under SSIM. Our method, as expected, deviates slightly more when compared to the baseline, but still achieves good performance, showcasing a high similarity between the

two images. The baseline is expected to have higher similarity since, by adding random Gaussian noise, it overall injects less information with respect to our method.

**Prediction performance** In Figure 4, we highlight the absolute mean difference in the prediction confidence between the corrupted and original image relative to the class of the original most confident prediction. Despite inducing a higher prediction change compared to the baseline, only high values of $\alpha$ reach a change greater than $10\%$ and are otherwise on average smaller than $5\%$. Once more, in the case of our approach, Vit-B16 is weaker than ResNet-18, resulting in lower changes in the predictions. This follows as a result of the architecture being more robust to input noise perturbation (Bhojanapalli et al., 2021; Shao et al., 2021). Interestingly enough, this robustness doesn't seem to transfer to the domain of explanations as previously highlighted in Figure 2. In Table 1 we

Table 1: Comparison of the different models and explainability tested, for the various values of $\alpha$ and top-$k$ used, including both the percentage of change of the explanation and the percentage drop in confidence of the model. For each cross-section of a value of $\alpha$ and a value of top-$k$ there are four values. In bold we highlight the values of our method, while the other represent the baseline. In both cases, the first number is the percentage of change in the explanation induced, while the smaller value, sided by the $\downarrow$, is the mean absolute prediction confidence change in the class predicted on the original image. Similarly, the values below them are the metric computed on the baseline.

| | $\alpha \setminus$top-$k$ | DeepLIFT SHAP | | | | | | |
|---|---|---|---|---|---|---|---|---|
| | | 0.01 | 0.05 | 0.1 | 0.2 | 0.4 | 0.6 | 0.8 |
| ResNet-18 | 0.03 | **11.5 0.33↓** | **27.2 0.74↓** | **37.6 1.06↓** | **49.0 1.45↓** | **59.9 1.96↓** | **64.7 2.27↓** | **67.4 2.47↓** |
| | | 15.1 0.14↓ | 29.1 0.29↓ | 35.4 0.43↓ | 41.9 0.62↓ | 48.8 0.88↓ | 51.8 1.04↓ | 53.5 1.19↓ |
| | 0.06 | **17.8 0.75↓** | **39.8 1.72↓** | **52.8 2.33↓** | **66.9 3.18↓** | **79.5 4.47↓** | **85.1 5.17↓** | **87.7 5.63↓** |
| | | 21.8 0.31↓ | 41.2 0.65↓ | 49.9 0.93↓ | 58.1 1.30↓ | 66.0 1.79↓ | 69.9 2.16↓ | 72.1 2.50↓ |
| | 0.09 | **22.9 1.21↓** | **48.4 2.66↓** | **63.0 3.62↓** | **78.2 5.04↓** | **91.1 7.14↓** | **96.3 8.42↓** | **98.9 9.04↓** |
| | | 27.1 0.50↓ | 50.4 1.04↓ | 59.8 1.48↓ | 69.0 2.05↓ | 77.2 2.94↓ | 81.2 3.48↓ | 83.4 3.93↓ |
| | 0.12 | **27.1 1.69↓** | **55.3 3.68↓** | **70.6 5.02↓** | **86.2 7.07↓** | **98.9 9.92↓** | **103.8 11.75↓** | **106.1 12.85↓** |
| | | 31.6 0.73↓ | 57.1 1.44↓ | 67.5 2.08↓ | 77.0 2.91↓ | 85.4 4.26↓ | 89.0 4.87↓ | 91.2 5.41↓ |
| | 0.15 | **30.9 2.16↓** | **61.1 4.74↓** | **76.6 6.51↓** | **92.2 9.10↓** | **104.0 12.82↓** | **108.8 15.29↓** | **110.6 16.76↓** |
| | | 35.7 0.97↓ | 63.1 1.85↓ | 73.6 2.68↓ | 83.0 3.81↓ | 90.8 5.59↓ | 94.4 6.27↓ | 96.4 6.92↓ |
| ViT-B16 | 0.03 | **25.9 0.34↓** | **54.2 0.77↓** | **71.9 1.08↓** | **88.6 1.44↓** | **105.8 1.88↓** | **115.6 2.14↓** | **120.5 2.25↓** |
| | | 11.0 0.06↓ | 23.2 0.13↓ | 31.5 0.18↓ | 41.1 0.29↓ | 52.1 0.41↓ | 56.8 0.50↓ | 59.0 0.54↓ |
| | 0.06 | **45.4 0.76↓** | **87.1 1.80↓** | **109.3 2.41↓** | **129.3 3.27↓** | **144.3 4.26↓** | **153.3 4.74↓** | **156.9 5.05↓** |
| | | 21.6 0.12↓ | 43.0 0.32↓ | 56.3 0.46↓ | 68.7 0.68↓ | 82.1 0.91↓ | 86.8 1.04↓ | 89.4 1.11↓ |
| | 0.09 | **60.8 1.21↓** | **109.2 2.82↓** | **130.6 3.86↓** | **149.0 5.14↓** | **164.1 6.51↓** | **168.3 7.28↓** | **172.1 7.81↓** |
| | | 30.8 0.21↓ | 58.5 0.53↓ | 73.6 0.75↓ | 85.6 1.08↓ | 98.7 1.40↓ | 103.1 1.53↓ | 105.4 1.69↓ |
| | 0.12 | **72.5 1.60↓** | **123.2 3.73↓** | **145.2 5.12↓** | **160.5 6.76↓** | **173.9 8.57↓** | **178.2 9.67↓** | **181.5 10.50↓** |
| | | 39.2 0.31↓ | 71.5 0.72↓ | 84.8 1.01↓ | 99.3 1.42↓ | 108.1 1.84↓ | 112.2 2.01↓ | 114.4 2.22↓ |
| | 0.15 | **78.7 1.97↓** | **133.1 4.46↓** | **153.7 6.12↓** | **168.7 8.09↓** | **181.9 10.39↓** | **185.1 11.86↓** | **188.8 12.97↓** |
| | | 47.2 0.41↓ | 80.6 0.89↓ | 94.5 1.25↓ | 105.3 1.72↓ | 113.2 2.24↓ | 118.8 2.44↓ | 122.0 2.74↓ |

report for some values of $\alpha$ and top-$k$, across different architectures, and for DeepLIFT Shap, the percentage of change in the explanation, and the percentage drop in prediction of our method and the baseline (the full table and other explainability methods are available at Table B.1, B.2, and B.3). This highlights that, especially as top-$k$ increases, our method induces a similar change in explanation and prediction, with a lower value of $\alpha$; this, by the method's structure, is less detectable as the weighted merge with a lower value. The table furthermore emphasizes that for the same values of $\alpha$ and top-$k$, the ViT-B16 is more vulnerable to our attack, with a higher explainability percentage change and a lower prediction drop. Lastly, we empirically validate the intuition behind choosing a picture from the running-up class, by which we refer to the class with the second-highest classification (or misclassifications in this case) by the model. In a real-world scenario, there are countless images from which to extract noise (injection) and insert it into the target image, especially in cases like ImageNet, where the number of classes is large. Figure 5, computed on the CIFAR-10 dataset,

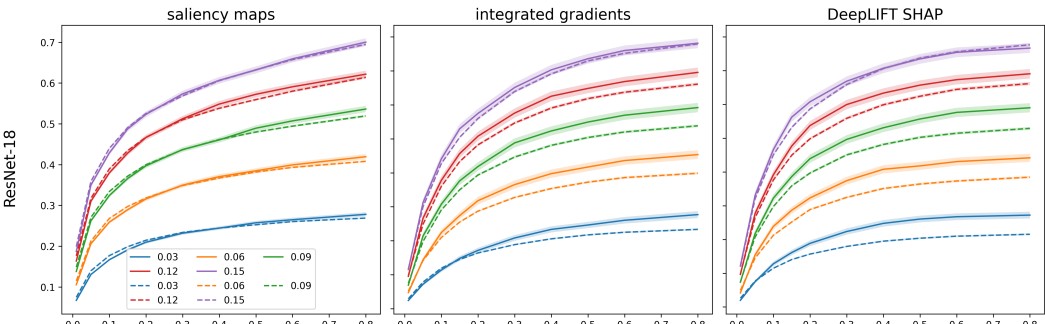

Figure 5: The figure compares the mean and standard deviation of injecting the original image with one from a running-up class (full line) vs the average of using all the other classes (dotted lines). The lines have been computed on the CIFAR10 dataset, with a ResNet-18 and different explainability methods (saliency maps, integrated gradients, and DeepLIFT SHAP). The computation scales linearly in the number of classes; therefore, it's not practical to compute the same graph on ImageNet.

shows that the explanation change induced by the images of the running-up class (full lines) is always as good as or better than picking an attack image from any other class. As intuition suggests, to induce the most 'confusion' in a model's explanations, you should inject features of an image of the class for which the model is mostly confused in the prediction.

## 5 CONCLUSIONS

We proposed a new black-box model-agnostic one-step adversarial attack on the explanations. The method exploits the uncertainties of the classification model to select an image from which to extract features that are injected into the original image. Our approach allows us to control how many features to inject and how much they should be weighted compared to the original image to obtain an undetectable attack. The attack hardly changes the prediction scores while significantly disrupting the original explanations. Our work serves as a fundamental step towards achieving more robust explainability models, especially in safety-critical domains like medicine, where XAI is used to understand diseases and illnesses, and the stakes are high. Future research directions can be grouped as follows:

- *robustness*: includes utilizing our new method to understand how the model robustness improves if fine-tuned on such attacks.

- *different domain*: currently, the attack has been created and tested on images. Other domains of interest are the signal domain (where, given the usually smaller input space, detecting alteration is easier), as well as the video domains

- *attack improvement*: one area that we did not explore in depth is the step where the image and injections are combined with a weighted sum. Although simple and effective, this approach can be improved by using more advanced blending techniques, such as local pixel-aware blending and Poisson blending, to reduce the detectability of the attack while maintaining the high explanation manipulation. Another area of improvement could be adapting the method to multi-label classification tasks. In this scenario, XAI methods usually produce as many attributions as classes. A possible adaptation could either create an image for each class or search for a common "running-up" class that was not recognized in the original prediction.

**Ethics statement** This paper proposes a novel adversarial attack on the explanation to spread awareness about this new kind of technique. We do not identify any violation.

**Reproducibility statement** The paper provides all the details of the method, including algorithm details and experimental settings, models, datasets, and parameters used. Implementation and code will be shared with the public if the paper gets accepted.

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

## A  APPENDIX: ON THE CHOICE OF THE MOST CONFIDENT CLASS PREDICTION IMAGE AS ATTACK IMAGE

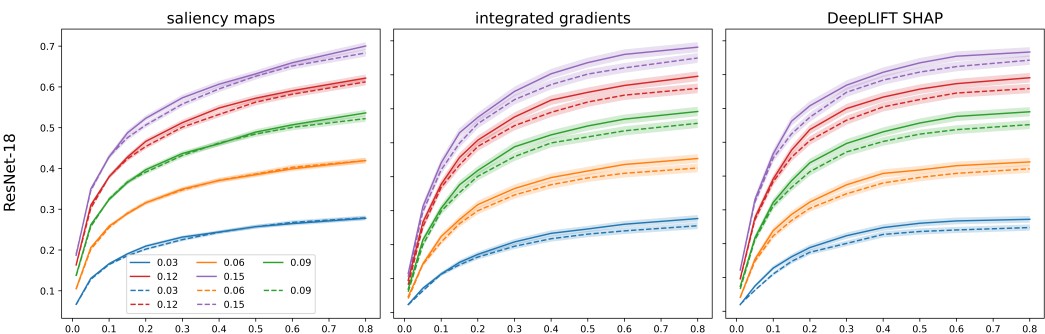

Figure A.1: Comparison between selecting the images with the highest confidence in the running-up class (full lines) vs choosing the images ranked 95-th to 100-th in confidence (dotted lines).

In the test we ran, to choose an image from a specific class to extract the explanations to inject into the original image, we selected the image with the highest confidence in the prediction of the running-up class. These images produce the strongest attribution explanations and therefore induce the highest result in the attack. Figure A.1, computed on the CIFAR-10 dataset, shows the difference in the attack performances between choosing the top 5 images in a class to perform the attack (full line) vs picking the images ranked 95-th to 100-th in the ordered confidence score for the running up class (dotted line). This result could also be inferred from Figure 5, since using a different class for attacking instead of the original image is like choosing a low confidence image in the running up class.

## B  APPENDIX: FULL TABLES OF PERFORMANCE COMPARISON

We now report the full tables that compare our algorithm and the baseline across all the $\alpha$, top-$k$, model architectures, and explainability methods tested.

Table B.1: Comparison of our method with the baseline in the case of Saliency maps

| | $\alpha$ \ top-$k$ | Saliency maps | | | | |
|---|---|---|---|---|---|---|
| | | 0.01 | 0.05 | 0.1 | 0.15 | 0.2 |
| ResNet-18 | 0.03 | **24.9** 0.37↓ 
 22.2 0.14↓ | **41.1** 0.80↓ 
 34.5 0.29↓ | **48.9** 1.09↓ 
 40.1 0.43↓ | **53.4** 1.33↓ 
 43.4 0.54↓ | **56.4** 1.47↓ 
 45.7 0.62↓ |
| | 0.06 | **34.0** 0.86↓ 
 30.3 0.31↓ | **52.9** 1.84↓ 
 45.7 0.65↓ | **61.3** 2.50↓ 
 52.2 0.93↓ | **65.9** 2.97↓ 
 55.7 1.18↓ | **68.9** 3.29↓ 
 58.3 1.30↓ |
| | 0.09 | **40.1** 1.35↓ 
 36.1 0.50↓ | **60.0** 2.89↓ 
 53.1 1.04↓ | **68.3** 3.94↓ 
 59.6 1.48↓ | **72.7** 4.80↓ 
 63.1 1.81↓ | **75.5** 5.35↓ 
 65.7 2.05↓ |
| | 0.12 | **44.8** 1.84↓ 
 40.7 0.73↓ | **64.9** 3.98↓ 
 58.2 1.44↓ | **72.8** 5.52↓ 
 64.8 2.08↓ | **76.8** 6.73↓ 
 68.1 2.55↓ | **79.3** 7.47↓ 
 70.5 2.91↓ |
| | 0.15 | **48.5** 2.36↓ 
 44.5 0.97↓ | **68.4** 5.16↓ 
 62.2 1.85↓ | **75.9** 7.09↓ 
 68.5 2.68↓ | **79.5** 8.54↓ 
 71.7 3.31↓ | **81.7** 9.55↓ 
 73.9 3.81↓ |
| Vit-B16 | 0.03 | **22.3** 0.30↓ 
 8.8 0.06↓ | **40.5** 0.61↓ 
 17.7 0.13↓ | **49.8** 0.75↓ 
 23.6 0.19↓ | **54.2** 0.84↓ 
 27.2 0.25↓ | **57.7** 0.93↓ 
 30.3 0.29↓ |
| | 0.06 | **37.2** 0.72↓ 
 16.6 0.12↓ | **64.1** 1.42↓ 
 31.5 0.32↓ | **75.4** 1.76↓ 
 40.7 0.46↓ | **81.5** 1.97↓ 
 45.7 0.59↓ | **84.7** 2.10↓ 
 49.2 0.68↓ |
| | 0.09 | **50.1** 1.19↓ 
 23.1 0.21↓ | **81.6** 2.38↓ 
 42.1 0.53↓ | **93.4** 2.94↓ 
 52.4 0.75↓ | **98.6** 3.26↓ 
 57.4 0.92↓ | **101.8** 3.48↓ 
 60.9 1.08↓ |
| | 0.12 | **59.2** 1.66↓ 
 28.8 0.31↓ | **93.5** 3.36↓ 
 51.2 0.72↓ | **103.4** 4.15↓ 
 60.1 1.01↓ | **109.2** 4.67↓ 
 66.0 1.22↓ | **112.1** 4.98↓ 
 70.5 1.42↓ |
| | 0.15 | **66.3** 2.12↓ 
 34.4 0.41↓ | **102.7** 4.32↓ 
 57.2 0.89↓ | **111.9** 5.35↓ 
 66.7 1.25↓ | **116.0** 6.00↓ 
 71.7 1.49↓ | **119.4** 6.45↓ 
 75.0 1.72↓ |

| | $\alpha$ \ top-$k$ | Saliency maps | | | | |
|---|---|---|---|---|---|---|
| | | 0.3 | 0.4 | 0.5 | 0.6 | 0.8 |
| ResNet-18 | 0.03 | **60.2** 1.74↓ 
 48.8 0.75↓ | **62.2** 1.92↓ 
 51.3 0.88↓ | **63.2** 1.99↓ 
 52.6 0.98↓ | **63.5** 2.07↓ 
 53.7 1.04↓ | **62.2** 2.09↓ 
 55.2 1.19↓ |
| | 0.06 | **72.4** 3.85↓ 
 61.8 1.60↓ | **74.4** 4.29↓ 
 63.8 1.79↓ | **75.6** 4.52↓ 
 65.2 2.04↓ | **76.1** 4.64↓ 
 66.5 2.16↓ | **75.7** 4.63↓ 
 68.1 2.50↓ |
| | 0.09 | **78.6** 6.16↓ 
 68.7 2.59↓ | **80.5** 6.88↓ 
 70.8 2.94↓ | **81.7** 7.21↓ 
 72.2 3.32↓ | **82.4** 7.42↓ 
 73.4 3.48↓ | **82.6** 7.35↓ 
 74.9 3.93↓ |
| | 0.12 | **82.2** 8.54↓ 
 73.4 3.67↓ | **84.0** 9.49↓ 
 75.3 4.26↓ | **84.9** 9.85↓ 
 76.6 4.72↓ | **85.9** 10.19↓ 
 77.5 4.87↓ | **86.7** 10.22↓ 
 78.9 5.41↓ |
| | 0.15 | **84.4** 11.02↓ 
 76.5 4.79↓ | **86.0** 12.25↓ 
 78.0 5.59↓ | **87.1** 12.74↓ 
 79.2 6.07↓ | **88.0** 13.25↓ 
 80.0 6.27↓ | **89.6** 13.54↓ 
 81.3 6.92↓ |
| Vit-B16 | 0.03 | **62.0** 1.01↓ 
 34.8 0.34↓ | **64.6** 1.05↓ 
 37.8 0.41↓ | **66.9** 1.10↓ 
 39.7 0.46↓ | **67.5** 1.13↓ 
 41.0 0.50↓ | **68.1** 1.13↓ 
 42.5 0.54↓ |
| | 0.06 | **91.4** 2.31↓ 
 55.4 0.80↓ | **94.8** 2.50↓ 
 58.5 0.91↓ | **98.4** 2.66↓ 
 62.1 0.99↓ | **99.7** 2.77↓ 
 61.9 1.04↓ | **105.1** 2.87↓ 
 64.0 1.12↓ |
| | 0.09 | **107.4** 3.84↓ 
 66.6 1.22↓ | **111.8** 4.10↓ 
 70.3 1.40↓ | **116.9** 4.34↓ 
 73.8 1.47↓ | **121.1** 4.57↓ 
 74.0 1.53↓ | **127.8** 4.87↓ 
 76.4 1.69↓ |
| | 0.12 | **118.2** 5.44↓ 
 73.6 1.62↓ | **121.6** 5.81↓ 
 78.0 1.84↓ | **127.0** 6.14↓ 
 79.7 1.93↓ | **132.2** 6.49↓ 
 81.9 2.01↓ | **140.8** 6.95↓ 
 84.3 2.22↓ |
| | 0.15 | **125.5** 7.02↓ 
 78.9 1.99↓ | **130.0** 7.55↓ 
 82.4 2.24↓ | **135.6** 7.95↓ 
 85.2 2.39↓ | **140.9** 8.52↓ 
 87.5 2.44↓ | **154.0** 9.26↓ 
 91.2 2.74↓ |

Table B.2: Comparison of our method with the baseline in the case of Integrated Gradients

| | Intergated Gradients | | | | |
|---|---|---|---|---|---|
| $\alpha$ \top-$k$ | 0.01 | 0.05 | 0.1 | 0.15 | 0.2 |
| **ResNet-18** | | | | | |
| 0.03 | **14.0** 0.36↓ 
 17.5 0.14↓ | **30.7** 0.76↓ 
 32.0 0.29↓ | **40.7** 1.04↓ 
 38.7 0.43↓ | **47.2** 1.26↓ 
 42.7 0.54↓ | **52.1** 1.41↓ 
 45.6 0.62↓ |
| 0.06 | **21.8** 0.77↓ 
 25.1 0.31↓ | **44.5** 1.63↓ 
 45.1 0.65↓ | **57.5** 2.34↓ 
 54.2 0.93↓ | **65.5** 2.80↓ 
 59.3 1.18↓ | **71.5** 3.18↓ 
 63.0 1.31↓ |
| 0.09 | **28.1** 1.22↓ 
 31.1 0.50↓ | **54.6** 2.58↓ 
 54.9 1.03↓ | **69.2** 3.69↓ 
 65.0 1.49↓ | **78.3** 4.50↓ 
 70.3 1.82↓ | **84.5** 5.08↓ 
 74.8 2.05↓ |
| 0.12 | **33.5** 1.68↓ 
 36.2 0.73↓ | **62.8** 3.60↓ 
 62.3 1.44↓ | **78.2** 5.10↓ 
 73.2 2.08↓ | **87.3** 6.12↓ 
 79.0 2.57↓ | **93.4** 6.99↓ 
 83.6 2.93↓ |
| 0.15 | **38.4** 2.17↓ 
 40.8 0.97↓ | **69.4** 4.63↓ 
 68.8 1.84↓ | **84.8** 6.47↓ 
 79.9 2.68↓ | **94.0** 7.83↓ 
 85.9 3.30↓ | **100.1** 9.07↓ 
 90.3 3.83↓ |
| **Vit-B16** | | | | | |
| 0.03 | **12.3** 0.32↓ 
 4.2 0.06↓ | **28.2** 0.81↓ 
 9.1 0.13↓ | **37.9** 1.13↓ 
 13.1 0.18↓ | **44.3** 1.36↓ 
 15.5 0.25↓ | **49.1** 1.50↓ 
 17.8 0.29↓ |
| 0.06 | **23.8** 0.80↓ 
 8.2 0.12↓ | **48.4** 1.92↓ 
 18.3 0.32↓ | **62.4** 2.64↓ 
 25.1 0.46↓ | **70.7** 3.08↓ 
 29.7 0.59↓ | **76.8** 3.46↓ 
 33.1 0.68↓ |
| 0.09 | **32.5** 1.28↓ 
 12.2 0.21↓ | **62.9** 3.07↓ 
 26.7 0.53↓ | **79.1** 4.20↓ 
 35.6 0.75↓ | **88.3** 4.95↓ 
 41.8 0.92↓ | **94.5** 5.49↓ 
 45.7 1.08↓ |
| 0.12 | **39.4** 1.72↓ 
 16.2 0.31↓ | **74.2** 4.11↓ 
 34.5 0.72↓ | **90.8** 5.62↓ 
 45.3 1.01↓ | **99.6** 6.60↓ 
 51.7 1.22↓ | **106.2** 7.36↓ 
 56.8 1.42↓ |
| 0.15 | **45.2** 2.09↓ 
 20.2 0.41↓ | **82.5** 4.98↓ 
 41.1 0.89↓ | **99.2** 6.81↓ 
 52.9 1.25↓ | **108.8** 8.07↓ 
 59.7 1.49↓ | **115.7** 9.03↓ 
 65.3 1.72↓ |

| | Intergated Gradients | | | | |
|---|---|---|---|---|---|
| $\alpha$ \top-$k$ | 0.3 | 0.4 | 0.5 | 0.6 | 0.8 |
| **ResNet-18** | | | | | |
| 0.03 | **58.9** 1.70↓ 
 49.6 0.75↓ | **63.3** 1.93↓ 
 53.0 0.88↓ | **66.3** 2.12↓ 
 54.7 0.98↓ | **68.6** 2.26↓ 
 56.1 1.04↓ | **71.6** 2.48↓ 
 57.8 1.18↓ |
| 0.06 | **79.7** 3.85↓ 
 68.3 1.60↓ | **84.9** 4.33↓ 
 71.4 1.79↓ | **88.6** 4.74↓ 
 73.6 2.04↓ | **91.1** 5.05↓ 
 75.6 2.15↓ | **94.2** 5.52↓ 
 77.8 2.49↓ |
| 0.09 | **93.0** 6.15↓ 
 80.1 2.58↓ | **98.3** 6.93↓ 
 83.8 2.94↓ | **101.7** 7.43↓ 
 86.2 3.29↓ | **104.1** 7.95↓ 
 87.9 3.46↓ | **107.3** 8.65↓ 
 90.2 3.92↓ |
| 0.12 | **102.0** 8.52↓ 
 88.8 3.67↓ | **106.9** 9.46↓ 
 92.6 4.23↓ | **110.3** 10.25↓ 
 94.9 4.68↓ | **112.5** 11.00↓ 
 96.5 4.84↓ | **115.2** 12.06↓ 
 98.7 5.40↓ |
| 0.15 | **108.4** 10.95↓ 
 95.6 4.75↓ | **113.1** 12.23↓ 
 98.9 5.57↓ | **116.4** 13.42↓ 
 101.1 6.02↓ | **118.5** 14.36↓ 
 102.4 6.23↓ | **120.9** 15.90↓ 
 104.5 6.90↓ |
| **Vit-B16** | | | | | |
| 0.03 | **55.8** 1.75↓ 
 21.1 0.34↓ | **60.8** 1.95↓ 
 23.6 0.41↓ | **64.3** 2.12↓ 
 25.3 0.46↓ | **66.7** 2.23↓ 
 26.7 0.50↓ | **69.5** 2.38↓ 
 28.5 0.54↓ |
| 0.06 | **85.3** 4.01↓ 
 38.5 0.80↓ | **90.7** 4.41↓ 
 42.4 0.91↓ | **94.2** 4.69↓ 
 45.4 0.99↓ | **97.1** 4.93↓ 
 47.8 1.04↓ | **100.4** 5.20↓ 
 50.0 1.11↓ |
| 0.09 | **102.7** 6.37↓ 
 52.7 1.22↓ | **108.0** 6.95↓ 
 57.2 1.40↓ | **111.7** 7.36↓ 
 60.4 1.47↓ | **114.3** 7.67↓ 
 62.4 1.53↓ | **117.9** 8.15↓ 
 65.1 1.69↓ |
| 0.12 | **114.6** 8.53↓ 
 63.3 1.62↓ | **119.9** 9.36↓ 
 68.2 1.84↓ | **123.7** 9.99↓ 
 71.4 1.93↓ | **126.7** 10.45↓ 
 73.8 2.01↓ | **130.8** 11.13↓ 
 76.8 2.22↓ |
| 0.15 | **123.8** 10.46↓ 
 72.1 1.99↓ | **129.4** 11.49↓ 
 77.0 2.24↓ | **134.1** 12.33↓ 
 80.0 2.39↓ | **137.0** 12.93↓ 
 82.8 2.44↓ | **140.8** 13.85↓ 
 85.9 2.74↓ |

Table B.3: Comparison of our method with the baseline in the case of DeepLIFT Shap

| | $\alpha$ \top-$k$ | DeepLIFT Shap | | | | |
| --- | --- | --- | --- | --- | --- | --- |
| | | 0.01 | 0.05 | 0.1 | 0.15 | 0.2 |
| ResNet-18 | 0.03 | **11.5** 0.33↓ | **27.2** 0.74↓ | **37.6** 1.06↓ | **44.4** 1.24↓ | **49.0** 1.45↓ |
| | | 15.1 0.14↓ | 29.1 0.29↓ | 35.4 0.43↓ | 39.2 0.54↓ | 41.9 0.62↓ |
| | 0.06 | **17.8** 0.75↓ | **39.8** 1.72↓ | **52.8** 2.33↓ | **61.0** 2.76↓ | **66.9** 3.18↓ |
| | | 21.8 0.31↓ | 41.2 0.65↓ | 49.9 0.93↓ | 54.5 1.18↓ | 58.1 1.30↓ |
| | 0.09 | **22.9** 1.21↓ | **48.4** 2.66↓ | **63.0** 3.62↓ | **72.1** 4.35↓ | **78.2** 5.04↓ |
| | | 27.1 0.50↓ | 50.4 1.04↓ | 59.8 1.48↓ | 64.9 1.81↓ | 69.0 2.05↓ |
| | 0.12 | **27.1** 1.69↓ | **55.3** 3.68↓ | **70.6** 5.02↓ | **79.9** 6.14↓ | **86.2** 7.07↓ |
| | | 31.6 0.73↓ | 57.1 1.44↓ | 67.5 2.08↓ | 73.0 2.55↓ | 77.0 2.91↓ |
| | 0.15 | **30.9** 2.16↓ | **61.1** 4.74↓ | **76.6** 6.51↓ | **86.0** 7.90↓ | **92.2** 9.10↓ |
| | | 35.7 0.97↓ | 63.1 1.85↓ | 73.6 2.68↓ | 79.2 3.31↓ | 83.0 3.81↓ |
| Vit-B16 | 0.03 | **25.9** 0.34↓ | **54.2** 0.77↓ | **71.9** 1.08↓ | **81.1** 1.26↓ | **88.6** 1.44↓ |
| | | 11.0 0.06↓ | 23.2 0.13↓ | 31.5 0.18↓ | 36.7 0.25↓ | 41.1 0.29↓ |
| | 0.06 | **45.4** 0.76↓ | **87.1** 1.80↓ | **109.3** 2.41↓ | **122.1** 2.88↓ | **129.3** 3.27↓ |
| | | 21.6 0.12↓ | 43.0 0.32↓ | 56.3 0.46↓ | 63.7 0.59↓ | 68.7 0.68↓ |
| | 0.09 | **60.8** 1.21↓ | **109.2** 2.82↓ | **130.6** 3.86↓ | **141.5** 4.60↓ | **149.0** 5.14↓ |
| | | 30.8 0.21↓ | 58.5 0.53↓ | 73.6 0.75↓ | 80.8 0.92↓ | 85.6 1.08↓ |
| | 0.12 | **72.5** 1.60↓ | **123.2** 3.73↓ | **145.2** 5.12↓ | **155.4** 6.06↓ | **160.5** 6.76↓ |
| | | 39.2 0.31↓ | 71.5 0.72↓ | 84.8 1.01↓ | 92.7 1.22↓ | 99.3 1.42↓ |
| | 0.15 | **78.7** 1.97↓ | **133.1** 4.46↓ | **153.7** 6.12↓ | **162.0** 7.22↓ | **168.7** 8.09↓ |
| | | 47.2 0.41↓ | 80.6 0.89↓ | 94.5 1.25↓ | 100.4 1.49↓ | 105.3 1.72↓ |

| | $\alpha$ \top-$k$ | DeepLIFT Shap | | | | |
| --- | --- | --- | --- | --- | --- | --- |
| | | 0.3 | 0.4 | 0.5 | 0.6 | 0.8 |
| ResNet-18 | 0.03 | **55.6** 1.74↓ | **59.9** 1.96↓ | **62.8** 2.11↓ | **64.7** 2.27↓ | **67.4** 2.47↓ |
| | | 45.7 0.75↓ | 48.8 0.88↓ | 50.6 0.98↓ | 51.8 1.04↓ | 53.5 1.19↓ |
| | 0.06 | **74.4** 3.95↓ | **79.5** 4.47↓ | **82.7** 4.82↓ | **85.1** 5.17↓ | **87.7** 5.63↓ |
| | | 63.1 1.60↓ | 66.0 1.79↓ | 68.2 2.04↓ | 69.9 2.16↓ | 72.1 2.50↓ |
| | 0.09 | **86.3** 6.29↓ | **91.1** 7.14↓ | **94.2** 7.74↓ | **96.3** 8.42↓ | **98.9** 9.04↓ |
| | | 74.0 2.59↓ | 77.2 2.94↓ | 79.7 3.32↓ | 81.2 3.48↓ | 83.4 3.93↓ |
| | 0.12 | **94.2** 8.72↓ | **98.9** 9.92↓ | **101.8** 10.83↓ | **103.8** 11.75↓ | **106.1** 12.85↓ |
| | | 81.8 3.67↓ | 85.4 4.26↓ | 87.4 4.72↓ | 89.0 4.87↓ | 91.2 5.41↓ |
| | 0.15 | **99.9** 11.19↓ | **104.0** 12.82↓ | **106.9** 14.11↓ | **108.8** 15.29↓ | **110.6** 16.76↓ |
| | | 88.1 4.79↓ | 90.8 5.59↓ | 92.9 6.07↓ | 94.4 6.27↓ | 96.4 6.92↓ |
| Vit-B16 | 0.03 | **98.7** 1.69↓ | **105.8** 1.88↓ | **112.1** 2.03↓ | **115.6** 2.14↓ | **120.5** 2.25↓ |
| | | 48.0 0.34↓ | 52.1 0.41↓ | 54.8 0.46↓ | 56.8 0.50↓ | 59.0 0.54↓ |
| | 0.06 | **138.2** 3.84↓ | **144.3** 4.26↓ | **150.1** 4.51↓ | **153.3** 4.74↓ | **156.9** 5.05↓ |
| | | 77.9 0.80↓ | 82.1 0.91↓ | 87.0 0.99↓ | 86.8 1.04↓ | 89.4 1.11↓ |
| | 0.09 | **157.7** 5.95↓ | **164.1** 6.51↓ | **167.1** 6.88↓ | **168.3** 7.28↓ | **172.1** 7.81↓ |
| | | 93.6 1.22↓ | 98.7 1.40↓ | 102.9 1.47↓ | 103.1 1.53↓ | 105.4 1.69↓ |
| | 0.12 | **168.0** 7.83↓ | **173.9** 8.57↓ | **176.9** 9.16↓ | **178.2** 9.67↓ | **181.5** 10.50↓ |
| | | 103.2 1.62↓ | 108.1 1.84↓ | 110.4 1.93↓ | 112.2 2.01↓ | 114.4 2.22↓ |
| | 0.15 | **176.1** 9.45↓ | **181.9** 10.39↓ | **184.4** 11.21↓ | **185.1** 11.86↓ | **188.8** 12.97↓ |
| | | 109.0 1.99↓ | 113.2 2.24↓ | 116.0 2.39↓ | 118.8 2.44↓ | 122.0 2.74↓ |

