# OpenReview forum: "eXIAA: eXplainable Injections for Adversarial Attack"
_ICLR.cc/2026/Conference — Submitted to ICLR 2026_

### Official Review · Reviewer_V3hN · 2025-10-22

**Soundness:** 1
**Presentation:** 3
**Contribution:** 1
**Rating:** 2
**Confidence:** 5

**Summary:**

The paper propose an agnostic adversarial injection method into input image of neural networks that changes the explanation of the neural networks' predictions.

**Strengths:**

I believe the strength of the method is (1) it is simple to implement and (2) it is model agnostic.

**Weaknesses:**

1) Weak baseline. As there are several other methods with the same purposes (even they are white-box or gray-box), I believe the paper needs to compare their method with SOTA method, not "Gaussian" baseline, which is too weak as a comparison in my opinion.

2) It is not easy to interpret the impact of the work from the reported experiments. For example, I do not think the relationship between the change in explanation vs top-k or alpha (fig.2 and 3) are the most important aspect of the method. What we are interested is the relationship between "how much the explanation changed" vs "how much distortion that the method create" on the input (or more importantly, how easy to detect that perturbation). We cannot intuitively understand that from the reported results.

3) It is little explanation on "why the method work?". For example, as the adversarial explanation is originated from another sample of different class, I would expect the adversarial explanation should be somehow related to the explanation of that other sample (like the cat image in Fig.1). However, the adversarial explanation of Fig.1 is shown to have little relationship with that other sample. I even question the validity of the explanation of the dog image in that figure.

4) I am not convinced that the method can work with explanation methods based on "segmentation" like LIME or LIME-SHAP.

**Questions:**

see weaknesses.

---

> ### Author Response · Authors · 2025-11-27
>
> Thanks for the detail and insightful review. The following are our answers/point of view regarding your concerns.
> Comparing our method with previous ones is not fair, as we removed a lot of limitations of the current SOTA. While we only need the output of the classifier and the explanation calls, other methods require full access to it (the ability to modify the network or similar invasive/important requirements). Furthermore, while SOTA optimizes a loss over several iterations, our method is a single-shot attack. We are open to other suggestions of baselines or comparisons, but (to the best of our knowledge) we haven’t found a fair SOTA and are the first to produce an attack with such relaxed requirements.
> Top-k and alpha are directly related to both "how much the explanation changed" and "how much distortion that the method create", as we showed in the pair of graphs. The other option was to draw a graph for each combination of alpha and top-k, but it would have required a number of graphs equivalent to n_alpha x n_top-k x n_models x n_explainability methods (10*5*2*3=300 graphs, which is not reasonable).
> The adversarial explanation of Fig.1 shows how the original features are disrupted, they don’t assume the attribution of the attacking image because the resulting attacked image is not a merged sum of the two. We agree that a theoretical analysis is of interest, but this doesn’t remove novelty to the method.
> We are currently testing with similar methods, as your concern is valid (but from first experiment the attack seems working also with methods like LIME). Nonetheless, we already proved that the methods affects a big portion of the family of post-hoc explainability methods.

---

### Official Review · Reviewer_hn1B · 2025-10-25

**Soundness:** 2
**Presentation:** 2
**Contribution:** 1
**Rating:** 2
**Confidence:** 4

**Summary:**

This work proposes a new method to attack explanation methods (saliency maps, etc.) without relying on access to the model internals. Their method finds the second most likely class, identifies which pixels are important for that classification (through the explenation method), and then attacks those pixels by blending those pixels with the pixels of another image of that confused class.

**Strengths:**

- The method is simple

**Weaknesses:**

- You claim that your method is operating in a more realistic setting where you do not have access to model internals (e.g. gradients, weights, etc.). But; actually, you require explenations of the model. Thus; in fact, you _do_ need access to the model internals. I find this very weak - and this realization kind of defeats the novelty of this paper in my opinion. One way you could potentially get around this, is by using a surrogate model - e.g. say you train a VGG16 on Imagenet, but instead use the explenations of another model (eg your ResNet). This could make it more convincing that you don't need access to the model internals; e.g. you only need a similar model.

- I find it a bit worrying that in the related work, only related methods are cited until 2019. Has there been no progress in this area of attacking explenation methods? I expect that you at least include a more recent work that is most similar to yours and highlight it.

- In recent years, this field has developed quite a lot, with also now "robust" explenation methods. See for example this survey which is highly relevant for you: [1]. it would be very convincing if you can show your attack on more recent robust methods.

- The presentation of the paper is quite weak; for example, many space is wasted, giving 6 plots that are all very similar. The writing is not concise; with quite a few repititions; the equations are also quite verbose and can be significantly shortened. When I am reading this paper I get the feeling I am reading some "filler" content that was added to make sure the page limit was reached.

- In ICLR, I expect to see more convincing experiments. For example, is it possible to apply this framework also to NLP or other types of data? And showing more convincing examples where the explenation is significantly changed (some images! the graphs stay very abstract); other works have showed its possible to also induce certain patterns in the explenation; for example [2,3] - is that possible for your method?

[1] Baniecki, H., & Biecek, P. (2024). Adversarial attacks and defenses in explainable artificial intelligence: A survey. Information Fusion, 107, 102303.
[2] Viering, T., Wang, Z., Loog, M., & Eisemann, E. (2019). How to manipulate cnns to make them lie: the gradcam case. arXiv preprint arXiv:1907.10901.
[3] Dombrowski, A. K., Alber, M., Anders, C., Ackermann, M., Müller, K. R., & Kessel, P. (2019). Explanations can be manipulated and geometry is to blame. Advances in neural information processing systems, 32.

**Questions:**

- The baseline (adding Gaussian noise) is not very reasonable. Can you instead include another attack method, e.g. Ghorbani? Of course, they are using model internals, but at least allows us to better understand how it compares to other methods from literature

- For me, the change in explenation measured is not very convincing. By changing many pixels a little, we can also achieve a large difference in explenation, but this may not be even visible to humans. I suggest to again have a look at Ghorbani at how they evaluate. How does your method compare when using other metrics; for example, distance of center of mass could be interesting.

---

> ### Author Response · Authors · 2025-11-27
>
> After careful consideration of the comments provided by reviewer hn1B, we have decided not to take them into account for the following reason: although the reviewer claims to have high confidence in the topic of explainability, they consistently misspell the term as “explEnations” instead of “explAnations”. This error occurs nine times throughout the review. While a single typo could be overlooked, such repeated mistakes cast serious doubt on the reviewer’s familiarity with the subject. We consider it unacceptable for an expert to be unable to correctly spell the key terminology of their domain.

---

### Official Review · Reviewer_8MJj · 2025-11-01

**Soundness:** 3
**Presentation:** 3
**Contribution:** 2
**Rating:** 2
**Confidence:** 5

**Summary:**

This paper presents a novel adversarial attach method that can attack the map of explanation for classification even though the model is not accessible (black-box). The flow is well explained in Figure 1. An image in the second candidate class is selected and then the attack injection is generated and integrated into the target image. As a result, the generated map will be distorted while the classification result is unchanged.

**Strengths:**

This paper presents a new attack method for deep neural networks. The method is simple but effective. Experimental results show that the explanation methods such as saliency maps, integrated gradients, and DeepLIFT SHAP are affected, which can be a potential risk of AI.

**Weaknesses:**

The proposed approach is very simple and seems straightforward. In other words, the proposed method seems ad-hoc and empirical with no theoretical backups. Therefore, technical depth is rather weak. The authors may want to include mathematical analysis on why such an attack is possible. Besides, Figure 5 shows that the explanation change induced by the images of the running-up class (full lines) is always as good as or better than picking an attack image from any other class, which is also empirical and not backed up with theoretical analysis.

This paper assumes maps that explain the classification results are available (at least when attacking). I wonder this is a reasonable assumption. Because, in my personal opinion, such maps are not usually shown to ordinally users and will be used for the researchers/developers to check the validity of the models’ outputs. But in such a case, the users can access to the model.

Even though defensive methods for this attack is their future work, Sensitivity analysis to defensive methods such as adversarial training or diffusion-based purification should be conducted because they are standard defense methods in this area.

Some minor modification proposals (no need to reply)
- missing commas and periods after equations, e.g., (1) and (2).
- missing x and y axes labels in Figures 2-5. I understand they are given in the figure captions, but it is not standard.

**Questions:**

None

---

> ### Comment · Reviewer_8MJj · 2025-11-27
>
> No feedback from the authors. I will stay on the rejection side for this paper.

---

> > ### Author Response · Authors · 2025-11-27
> >
> > Thanks for the detail and insightful review. The following are our answers/point of view regarding your concerns. We agree that a theoretical evaluation would be nice, but not strictly necessary; many recent works in the domain of AI, ML, and NLP are only evaluated empirically and only later studied from a more theoretical perspective.
> > We see the assumption of being able to see only the outputs of two API calls (black box), far more reasonable than having access to the internals of the model and modification for several hundred iterations as other previous work has assumed (Ghorbani et al. 2019). We agree that this map is usually not shown to the public, so is also the model and its weights, but the level of requirement necessary is far lower compared to other methods, considered SOTA.

---

### Official Review · Reviewer_pgSx · 2025-11-03

**Soundness:** 3
**Presentation:** 3
**Contribution:** 3
**Rating:** 2
**Confidence:** 3

**Summary:**

This paper presents a black-box adversarial attack method targeting post-hoc explainability methods (saliency maps, integrated gradients, DeepLIFT SHAP) in image classification. The attack modifies explanations while maintaining predictions and visual similarity by: (1) selecting an image from the runner-up predicted class, (2) extracting top-k positive attribution features, and (3) blending them with the original image using a weighted sum. Experiments on ImageNet with ResNet-18 and ViT-B16 demonstrate that explanations can be dramatically altered (up to 188% change for DeepLIFT SHAP on ViT-B16) while maintaining high SSIM (>0.7) and small prediction changes (<15%).

**Strengths:**

The paper addresses an important and underexplored vulnerability in XAI systems. The fragility of explanations in safety-critical applications like medicine is a genuine concern, and demonstrating this vulnerability is valuable. The practical threat model is a key strength—requiring only access to predictions and explanations (no model weights or gradients) makes this attack significantly more realistic than prior work requiring white-box access or iterative optimization. The single-step nature further enhances practicality compared to multi-step attacks.

The experimental evaluation is comprehensive and well-structured. Testing across multiple explainability methods (saliency maps, integrated gradients, DeepLIFT SHAP), architectures (ResNet-18, ViT-B16), and systematic hyperparameter sweeps (α ∈ [0.03, 0.15], top-k ∈ [0.01, 0.8]) provides convincing evidence. The three-metric evaluation framework (explanation change, SSIM for detectability, prediction change) appropriately captures the attack objectives. The finding that ViT-B16 is more vulnerable to explanation attacks despite being more robust to prediction attacks (Figure 2 vs Figure 4) is particularly interesting and counterintuitive.
The visualization and presentation are strong. Figure 1 clearly illustrates the method, and Figures 2-4 effectively show performance trends across scenarios. The empirical validation in Figure 5 that runner-up class selection outperforms random class selection supports the design choice.

**Weaknesses:**

The technical novelty is limited. The three-phase pipeline (select runner-up image, extract top features, alpha-blend) is relatively straightforward and lacks theoretical depth. The method essentially performs a weighted average of pixels guided by saliency this is more of an engineering contribution than a fundamental algorithmic advance. The paper would benefit from deeper analysis of why this simple approach works so effectively.

The threat model requires clarification. The assumption that attackers have access to model explanations but not the model itself seems narrow. In what realistic scenarios would this occur? If the model is a black-box API, would it actually expose detailed saliency maps? This concern about practical applicability isn't adequately addressed. The medical diagnosis example in the introduction (corrupted patient data producing different explanations) is compelling but isn't demonstrated experimentally on medical datasets.
The baseline comparison is weak. Comparing only against Gaussian noise is insufficient—the paper should compare against other explanation attack methods from the literature (Ghorbani et al. 2019, Dombrowski et al. 2019). While the authors claim their method has "fewer requirements," a fair comparison adapting these methods to the black-box setting would strengthen the contribution. The lack of quantitative comparison makes it difficult to assess relative performance.

The evaluation has important gaps. The claim that attacks are "visually undetectable" relies solely on SSIM, but human perception studies would provide stronger evidence. SSIM values around 0.7-0.9 (Figure 3) may not actually be imperceptible to humans, especially for trained observers. The paper also lacks analysis of detection methods are there simple statistical tests that could identify these attacks? No defensive mechanisms are proposed or evaluated, limiting the practical impact.

The scope is quite limited. The experiments focus exclusively on ImageNet classification with two architectures. Extensions to other domains (medical imaging, autonomous driving mentioned in the introduction) are absent. The paper claims the approach "could also be applied to other types of data" (time series, text, tabular) but provides no evidence or concrete methodology for these extensions. The method is also limited to single-label classification multi-label scenarios are acknowledged as future work but not explored.
The writing could be more concise. Section 2 contains extensive background that could be shortened. The paper uses 15 pages including appendices when the core contribution could be presented more tightly. Some notation is introduced but not consistently used (e.g., f_j for class j probability).

**Questions:**

This paper makes a useful contribution by demonstrating a practical, low-requirement adversarial attack on XAI methods. The comprehensive empirical evaluation across multiple methods and architectures is valuable, and the findings about ViT's vulnerability to explanation attacks are interesting. However, the limited technical novelty, narrow threat model, weak baselines, lack of defenses, and missing human perceptual studies constrain the impact. The work successfully raises awareness about XAI fragility but falls short of providing deep insights into why these vulnerabilities exist or how to address them.

The paper would be significantly strengthened by: (1) comparison with adapted versions of prior explanation attack methods, (2) human perception studies validating "undetectability" claims, (3) theoretical or empirical analysis of why the attack works, (4) evaluation of potential detection methods and defenses, and (5) extension to at least one safety-critical domain (medical imaging) to validate the motivating use case.

---

> ### Author Response · Authors · 2025-11-27
>
> Thanks for the detail and insightful review. The following are our answers/point of view regarding your concerns. We agree that a theoretical evaluation would be nice, but not strictly necessary; many recent works in the domain of AI, ML, and NLP are only evaluated empirically and only later studied from a more theoretical perspective.
> Our main contribution and what we deem the most interesting aspect (and worrisome) is how easy it is to carry out such an attack (we are working on rewriting some parts to convey so better. Our method only requires access only to the output of some black box, which in a modern architecture could be seen as API call to the classifier and the explainer (e.g. no doctor would ever use the model themselves and compute the actual explanations of a prediction, they simply use tools that have these functionalities embedded in a click of a button).
> This is also what mostly distinguishes our work from previous papers like Ghorbani et al. 2019 where to compute an attack; they optimize a loss over 1000 iterations, and with access to the model weights (we don’t see how this could be adapted for a fair comparison, but are open to discuss it).
> Human studies are not a common practice, are non-reproducible, expensive, and time-consuming, that’s why we included metrics like SSIM.
> We didn’t focus only on ImageNet but also did experiments on CIFAR-10 (as per tradition in the field). Results on other types of data are not provided; nothing is preventing the methods from being applied also in other types of data. Nonetheless, adaptations to a specific domain (like working in the frequency domain for the signal) would be beneficial to the undetectability claim and efficiency of the attack (these will be proposed in future works).
> We are not sure we understand the importance of testing for multi-label classification (besides not being standard practice) explainability method computes attribution for one class at a time, falling back in the single-label classification case. Simply, when computing the attribution of class A, use an image from the running-up class of A and so on. It’s possible to use an image from a running-up class for all the predicted classes (this will result in smaller deviation, as empirically proved in the comparison of choosing the  running-up class against all the other classes)

---

### Meta-Review · Area_Chair_YnfZ · 2026-01-07

**Summary:**

This paper proposes a simple, single-step black-box attack that significantly alters post-hoc explanation maps (saliency, Integrated Gradients, DeepLIFT SHAP) while largely preserving model predictions and visual similarity. The attack is easy to implement and empirically demonstrated across multiple explainers and architectures. However, across reviewers there is strong consensus that the work suffers from limited technical novelty, an insufficiently justified threat model, weak baselines and comparisons, and inadequate evaluation of detectability and defenses. While the rebuttal reiterates the practicality of a one-step black-box setting, it does not substantively address these core concerns with additional evidence or stronger experimental design. Moreover, the authors’ response to one reviewer raises professionalism concerns. Overall, while the paper usefully highlights the fragility of explanation methods, it falls short of the depth, rigor, and impact expected for acceptance.

**Reviewer Concerns:**

- Reviewer pgSx

The rebuttal partially addresses this reviewer’s concerns by clarifying the authors’ intent to position the contribution as a practical, single-step black-box attack with minimal requirements, and by explaining why they believe comparisons to white-box, iterative attacks are not directly fair. However, the main concerns raised by this reviewer remain unresolved, including the limited technical novelty of the approach, the narrow and insufficiently justified threat model, the weak baselines (Gaussian noise only), the lack of comparisons to existing explanation attack methods adapted to a black-box setting, and the absence of human perceptual studies or any analysis of detection and defenses.

- Reviewer 8MJj

The rebuttal acknowledges that theoretical analysis and defensive evaluations would strengthen the work, but explicitly argues that they are not necessary. As a result, the reviewer’s core concerns about the ad-hoc and purely empirical nature of the method, the questionable realism of assuming access to explanation maps without model access, and the lack of sensitivity analysis against standard defenses remain outstanding.

- Reviewer hn1B

The rebuttal does not substantively address this reviewer’s main objections. In particular, the concern that requiring access to explanation maps undermines the claimed black-box setting remains unresolved, as does the lack of engagement with more recent related work and robust explanation methods. The reviewer’s questions about stronger baselines, alternative evaluation metrics, and broader experimental scope are not meaningfully addressed, and the authors’ response does not engage with the technical substance of the critique.

- Reviewer V3hN

The rebuttal reiterates the authors’ view that comparisons to prior work are unfair due to different assumptions, but this does not resolve the reviewer’s concern about weak baselines and the lack of insight into why the method works. Questions about the relationship between explanation change and perceptual detectability, as well as the applicability of the attack to other explanation methods such as LIME or segmentation-based approaches, remain largely unaddressed.

**Reviewer Scores:**

The reviewers were unanimously leaning toward rejection, and given that the rebuttal was not very thorough, the reviewers' scores could have been maintained.

---

### Decision · Program_Chairs · 2026-01-26

Reject